# Creating boundaries along a synthetic frequency dimension

Avik Dutt [1,2], Luqi Yuan [3], Ki Youl Yang [1], Kai Wang [1], Siddharth Buddhiraju [1], Jelena Vučković [1] & Shanhui Fan [1 ✉]

Synthetic dimensions have garnered widespread interest for implementing high dimensional classical and quantum dynamics on low-dimensional geometries. Synthetic frequency dimensions, in particular, have been used to experimentally realize a plethora of bulk physics effects. However, in synthetic frequency dimension there has not been a demonstration of a boundary which is of paramount importance in topological physics due to the bulk-edge correspondence. Here we construct boundaries in the frequency dimension of dynamically modulated ring resonators by strongly coupling an auxiliary ring. We explore various effects associated with such boundaries, including confinement of the spectrum of light, discretization of the band structure, and the interaction of boundaries with one-way chiral modes in a quantum Hall ladder, which exhibits topologically robust spectral transport. Our demonstration of sharp boundaries fundamentally expands the capability of exploring topological physics, and has applications in classical and quantum information processing in synthetic frequency dimensions.

[1] Ginzton Laboratory and Department of Electrical Engineering, Stanford University, Stanford, CA 94305, USA. [2] Department of Mechanical Engineering, Institute for Physical Science and Technology, University of Maryland, College Park, MD 20742, USA. [3] State Key Laboratory of Advanced Optical Communication Systems and Networks, School of Physics and Astronomy, Shanghai Jiao Tong University, Shanghai 200240, China. ✉email: shanhui@stanford.edu

The concept of synthetic dimensions[1–3], whereby various degrees of freedom of atoms or photons are used to mimic spatial dimensions, is of significant recent interest for simulating high-dimensional phenomena on systems with fewer geometric dimensions. Synthetic dimensions have been formed by coupling states labeled by degrees of freedom such as spin[1,4], frequency[5,6], orbital angular momentum (OAM)[7], time bins[8–10] or transverse spatial supermodes[11]. Many interesting physical effects, including nontrivial topological phenomena and effective gauge fields for neutral ultracold atoms or photons, have been realized in synthetic dimensions.

Specifically for topological phenomena, constructing a sharp boundary in the synthetic dimension is of central importance. An essential concept in topological physics is the bulk-edge correspondence, which relates the existence and properties of edge modes in a finite lattice to the quantized topological invariant of the corresponding bulk (infinite) lattice. For Hermitian systems, examples of bulk-edge correspondence include the one-way chiral edge states at the boundary of a Chern insulator[12], the zero-energy edge modes of a Su-Schrieffer-Heeger model[13], and the recently discovered corner modes of a higher-order topological insulator[14–16]. Moreover, the bulk-edge correspondence has also been generalized to non-Hermitian systems, leading to intriguing phenomena such as the non-Hermitian skin effect[17–20]. Creating a boundary in the synthetic dimension is essential for further exploration of such phenomena in synthetic space. In addition, the creation of boundaries in synthetic dimensions is important for applications such as implementing arbitrary linear transformations for frequency conversion, quantum circuits, and photonic neural networks[21].

A prominent approach to create synthetic dimensions is to use the frequency modes of a ring resonator. Synthetic frequency dimensions have enabled experimental demonstrations of a plethora of bulk physical effects. For Hermitian systems, examples of these effects include Bloch oscillations[22–25], effective electric and magnetic gauge fields[26–30], spin-orbit coupling and consequent spin-momentum locking[27], complex long-range coupling[31,32], and chiral currents originating from the nontrivial topology of the quantum Hall effect[27]. For non-Hermitian systems, nontrivial eigenvalue topology such as topological winding or braiding of the energy bands have also been recently observed in frequency dimensions[33,34]. However, experimentally probing the edge implications of these bulk topological phenomena has remained an open challenge in synthetic frequency dimensions. Unlike systems in real space, synthetic lattices created using frequency modes typically do not have a well-defined boundary. In the absence of boundaries or defects, the robustness of light transport[35], which is one of the hallmarks of topological phenomena, has not been observed along the frequency axis.

In this paper, we provide an experimental demonstration for constructing boundaries in synthetic frequency dimensions. Previous theoretical works have investigated synthetic-space boundary effects by assuming sharp[5] or gradual[36] changes in the group-velocity dispersion of the waveguide forming the ring resonator, by strongly coupling an auxiliary ring[21], or by including memory elements[37]. Here we experimentally realize the approach of coupling to auxiliary ring resonators. We observe that an excitation within the finite lattice stays confined between the boundaries in synthetic space, resulting in the discretization of the band structure in reciprocal space. We also implement boundaries in a synthetic quantum Hall ladder geometry and demonstrate one-way propagation of topological chiral edge states that are immune to back reflection despite the presence of a boundary, thus constituting an observation of topologically robust transport of light along the frequency axis. With the added functionality of creating sharp edges, we anticipate the observation of higher-dimensional boundary phenomena that have been beyond the purview of real-space or synthetic-space topological photonics.

## Results

**Creation of boundaries in one dimension.** Consider a single ring resonator of length $L_0$ made of a waveguide with group velocity $v_g$ (Fig. 1a). In the absence of group velocity dispersion, the ring supports cavity modes equispaced in frequency by the free-spectral range (FSR) $\Omega_R = 2\pi v_g/L_0$. To excite these modes we couple the ring with an external waveguide at an amplitude coupling ratio $\gamma_0$. The resulting transmission spectrum, assuming that all the ring modes are critically coupled with an internal loss rate equal to the external coupling loss rate, is shown in Fig. 1c. The spectrum features a periodic array of resonant dips equally spaced by the FSR. These modes can be coupled to form a one-dimensional (1D) synthetic frequency lattice by electro-optically modulating the refractive index of a small portion of the ring at a modulation frequency $\Omega_M = \Omega_R$[5,6,32]. The Hamiltonian for such a system is[32,38],

$$H = J \sum_{m=-M}^{M} b_m^\dagger b_{m+1} + \text{H.c.} \tag{1}$$

where $b_m (b_m^\dagger)$ is the annihilation (creation) operator for a mode at frequency $\omega_m = \omega_0 + m\Omega_R$. For a single ring with $\omega_0 \gg \Omega_R$, a very large number of modes ($M > 100$) can be coupled along the synthetic frequency dimension, as demonstrated experimentally in refs. [25,32]. Thus a single modulated ring closely approximates the bulk behavior ($M \to \infty$) of a lattice.

To truncate such a lattice and create boundaries, we couple an auxiliary ring resonator of a smaller length $L_a < L_0$, corresponding to a larger FSR $\Omega_{R,a} = 2\pi v_g/L_a$ (Fig. 1b). Here we have assumed that the auxiliary ring is made of a waveguide with the same group velocity as the main ring, and is coupled to the main ring via a directional coupler with an amplitude coupling coefficient $\gamma_a$. Note that similar geometries have previously been used for optical communications, flat-band lattices, reconfigurable frequency conversion, and demonstrating coupled-resonator induced transparency[39–45].

As an illustration, Fig. 1d shows the spectral positions of the main cavity and auxiliary ring modes for $N = L_0/L_a = 6$ in the absence of modulation. The corresponding transmission spectrum is plotted in Fig. 1e. Near the frequencies where the resonances from the two rings align, if $\gamma_a > \gamma_0^2/2$, a splitting is induced (Fig. 1f). Here $\gamma_0^2$ is the power splitting ratio of the directional coupler between the input-output waveguide and the main ring. Unlike the spectrum in Fig. 1c, the spectrum here in Fig. 1e is no longer periodic with respect to translation by $\Omega_R$ along the frequency axis.

When the modulation is again introduced in the main ring with a modulation frequency $\Omega_M = \Omega_R$, the modulation can induce the transition between some of the modes. Specifically in Fig. 1e, the green arrows represent the allowed modulation-induced couplings along the frequency dimension, whereas the red crosses represent inhibition of the coupling to modes that are perturbed by the auxiliary ring. A series of several finite lattices are formed, which are separated by the split resonances induced by the auxiliary ring. The presence of the auxiliary ring thus can introduce a sharp boundary in the synthetic dimension.

**Characterization of the unmodulated resonators.** To experimentally characterize the resonator in the absence of modulation, we measure the transmission spectra (Fig. 2) in an experimental realization of the setup shown in Fig. 1a. The details of the experiments, which are implemented using fiber rings, are provided in Supplementary Information Section I. Without the

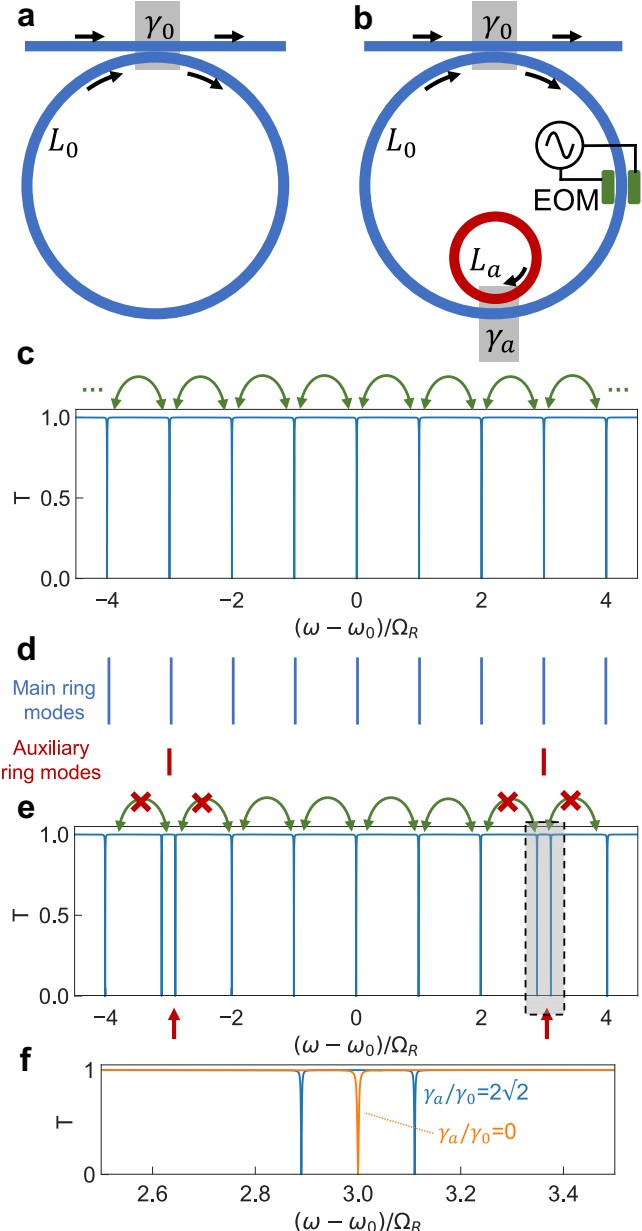

**Fig. 1 Ring resonator and its transmission spectrum with and without an auxiliary ring.** Schematic of a static ring resonator (**a**) and its simulated transmission $T$ (**c**). The frequencies of the ring's modes are indicated in (**d**) with blue lines. **b** Ring with a coupled auxiliary resonator (red), and its corresponding transmission spectrum in the absence of modulation (**e**). The frequencies of the auxiliary ring's modes are indicated in (**d**) with red lines. **f** Shows a zoom-in around the mode of the auxiliary resonator that is aligned to a mode of the main ring. **c**, **e**, **f** are calculated numerically using a scattering matrix method with a power splitting ratio of $\gamma_0^2 = 0.01$. For illustrative purposes, a propagation loss rate $\alpha_0$ in the main ring is chosen to critically couple it to the waveguide, $\exp(-\alpha_0 L_0) = 1 - \gamma_0^2 = 0.99$. The auxiliary ring is assumed to be lossless. The red crosses in (**e**) indicate that the modulation at the FSR cannot couple the split modes to the rest of the lattice as they are not aligned to the frequency grid of the main ring. EOM electro-optic modulator.

auxiliary ring, the transmission features a set of resonant dips, with minimum transmission $T_{\min} \approx 0.7$ that are similar for all the dips. These dips correspond to the resonances of the main ring. The frequency spacing of the nearest resonances as a

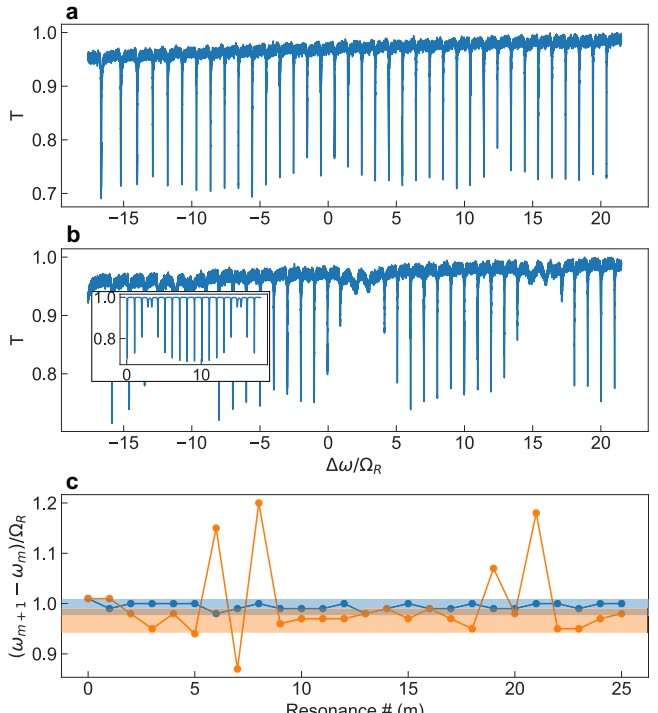

**Fig. 2 Measurement of transmission through a static ring resonator and frequency separation of modes.** Transmission spectrum without (**a**) and with (**b**) an auxiliary ring resonator coupled. $\Omega_R = 5.35$ MHz in (**a**), (**c**), $\Omega_R = 5.25$ MHz in (**b**). $L_0 = 38.6$ m, $L_0/L_a \approx 12$, $\gamma_0 = 0.1$, $\gamma_a/\gamma_0 = 5$. Inset in (**b**) shows a numerical calculation of the transmission spectrum based on a scattering matrix model, similar to Fig. 1d, but with finite roundtrip losses in both the main ring and the auxiliary ring of 5%. **c** Frequency difference between adjacent resonances of the main ring without an auxiliary ring from (**a**) (blue), and with an auxiliary ring from (**b**) (orange), as a function of the order of the resonances.

function of order of resonances is plotted as the blue line in Fig. 2c. We see that the frequency spacing is nearly a constant. In the presence of coupling to the auxiliary ring, there is a marked increase in $T_{\min}$ near the main cavity modes that are aligned to the auxiliary ring modes (Fig. 2b). The increase in $T_{\min}$ is in accordance with scattering matrix simulations including a loss in the auxiliary ring (inset of Fig. 2b), and this loss was ignored in Fig. 1d, e for simplicity. Around the resonant frequencies of the auxiliary ring, we see that the resonances of the coupled system are no longer equally spaced (orange line in Fig. 2c). In addition, for the coupled system, the frequency spacings between modes far away from the resonances of the auxiliary ring, which we define as the FSR of our coupled ring system, is smaller as compared to the FSR of the main ring by itself (Fig. 2c, see Supplementary Section II for an analytical derivation of this effect).

**Measurement of boundary effects in 1D lattice space.** For the remainder of the paper, we will consider a modulated resonator. We first demonstrate the effect of a boundary created by the auxiliary ring by measuring the steady-state intensity distribution in the synthetic frequency dimension (Fig. 3) in the presence of modulation. We excite the system at a frequency $\omega_{\text{in}}$ near one of the resonances of the main ring, the order of which is denoted by $m_0$. $\omega_{\text{in}}$ is gradually swept, and the detuning $\Delta\omega = \omega_{\text{in}} - \omega_{m_0}$ forms the vertical axis in Fig. 3a, b, d–f. At each input frequency, the frequency-lattice distribution of the steady-state cavity field

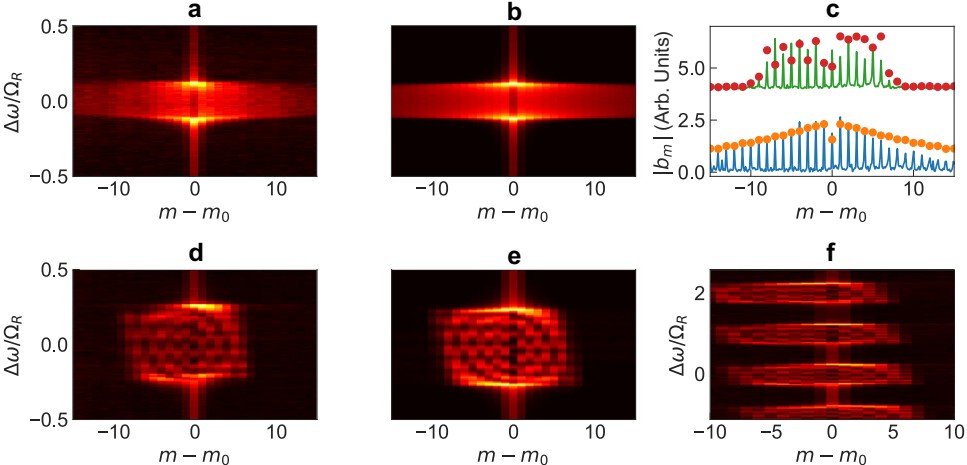

**Fig. 3 Amplitude distribution in frequency lattice space. a**, **b** Without and (**d**–**f**) with an auxiliary ring, corresponding to infinite and finite lattices respectively. **a**, **d**, **f** are experimentally measured heterodyne spectra. **b**, **e** are obtained from simulations based on a Floquet scattering matrix analysis. **c** Blue (infinite) and green (finite) curves represent line cuts through the raw heterodyne data at $\Delta\omega \approx 0$. Dots represent line cuts through respective simulated spectra in panels (**b**, **e**) respectively. The infinite lattice data without an auxiliary ring is vertically offset by 4 units. **f** The heterodyne spectra similar to (**d**) but over a much larger range of input laser detuning $\Delta\omega > \Omega_R$, thus exciting various lattice sites between the two boundaries. The breakdown of discrete modal translational symmetry is evident, as the response changes depending on which frequency site is excited. Due to reflection from the boundaries, fringes are visible in (**d**–**f**). Here the frequency mode axis ($m - m_0$) is measured with respect to the input laser frequency. Arb. Units arbitrary units.

is obtained from a heterodyne measurement of the transmitted field[46]. This frequency sideband number is denoted by $m - m_0$ along the horizontal axis in Fig. 3.

In the absence of the auxiliary ring, the transmitted field contains a large number of sidebands (Fig. 3a). This experimental data matches well with the simulated spectrum in Fig. 3b which was calculated using a Floquet scattering matrix analysis. The steady-state field intensity of the $m$-th sideband away from the input falls off exponentially as $\sim\exp(-|m - m_0|/\tau_p J)$ (see Fig. 3c bottom), for large $m - m_0$[25], where $\tau_p$ and $J$ are the ring photon lifetime and the modulation strength respectively.

On the other hand, when the auxiliary ring is coupled to the main ring, the output field contains a far smaller number of sidebands. This indicates that within the ring, the only modes excited are those that lie between the two boundaries along the frequency axis (experiment: Fig. 3d, simulations: Fig. 3e). We also observe interference fringes created by reflections from the boundaries. Note that the strengths of the fringes increase with an increase in the modulation-induced coupling strength, since light is able to traverse along the frequency axis for longer distances before getting dissipated. However, the strong confinement of light to within the boundaries is preserved as long as the splitting induced by the auxiliary ring resonator is larger than $2J$. Figure 3f illustrates the spectra upon exciting various lattice sites within the two boundaries. This result was obtained by sweeping the input laser detuning over a large range $\Delta\omega \gg \Omega_R$. Since the measured heterodyne spectrum is always referenced to the input laser frequency mode $m_0$, we observe a shift of the output spectrum towards lower frequency sidebands as $m_0$ increases.

**Measurement of 1D boundary effects in reciprocal space**. An infinite lattice that obeys discrete translational symmetry can be characterized by a conserved continuous quantum number, the Bloch quasimomentum $k \in [0, 2\pi)$, which labels the bulk properties in reciprocal space. For each $k$, one or more continuous bands are formed which correspond to the eigenenergy spectrum of the infinite lattice. In the frequency synthetic dimension, the wavevector along the frequency axis corresponds to a time variable. We have previously demonstrated a synthetic-space band

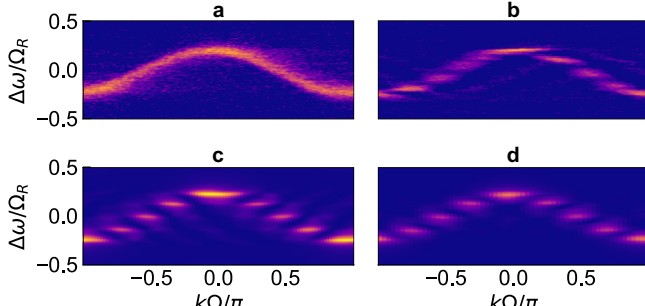

**Fig. 4 Band structure of bulk and finite lattices in one dimension. a** Measured band structure for a bulk lattice without any boundary, that is, without coupling to an auxiliary ring. A continuous band is observed. **b** Experimentally measured band structure from the time-resolved transmission of the main ring, when an auxiliary ring is coupled. A discrete band structure is seen, due to the effect of a boundary creating a finite lattice. **c** Floquet simulations using a full scattering matrix analysis of the structure in Fig. 1a, showing agreement with the experimental measurements in (**b**). **d** Result of a tight-binding Floquet analysis of Eq. (1), for a finite lattice with $M = 4$ (9 sites). The agreement of (**d**), which is not based on a modulated ring system but a general tight-binding lattice, establishes that a boundary can be realized along the synthetic frequency dimension using an auxiliary ring. $J/\Omega_R = 0.12$.

structure spectroscopy technique[32]. In this technique, we scan the input frequency of a continuous-wave laser. For each frequency, after the transient dissipates, we measure the transmission intensity as a function of time. Since the time corresponds to the wavevector $k$ along the synthetic frequency dimension, the resulting two-dimensional plot of transmission as a function of frequency and wavevector then provides a measure of the bandstructure. An example of such a measurement, for our system in the absence of the auxiliary ring, is shown in Fig. 4a. The locations of the peaks in the frequency-wavevector space closely match the band structure of a one-dimensional tight-binding model with nearest-neighbor coupling.

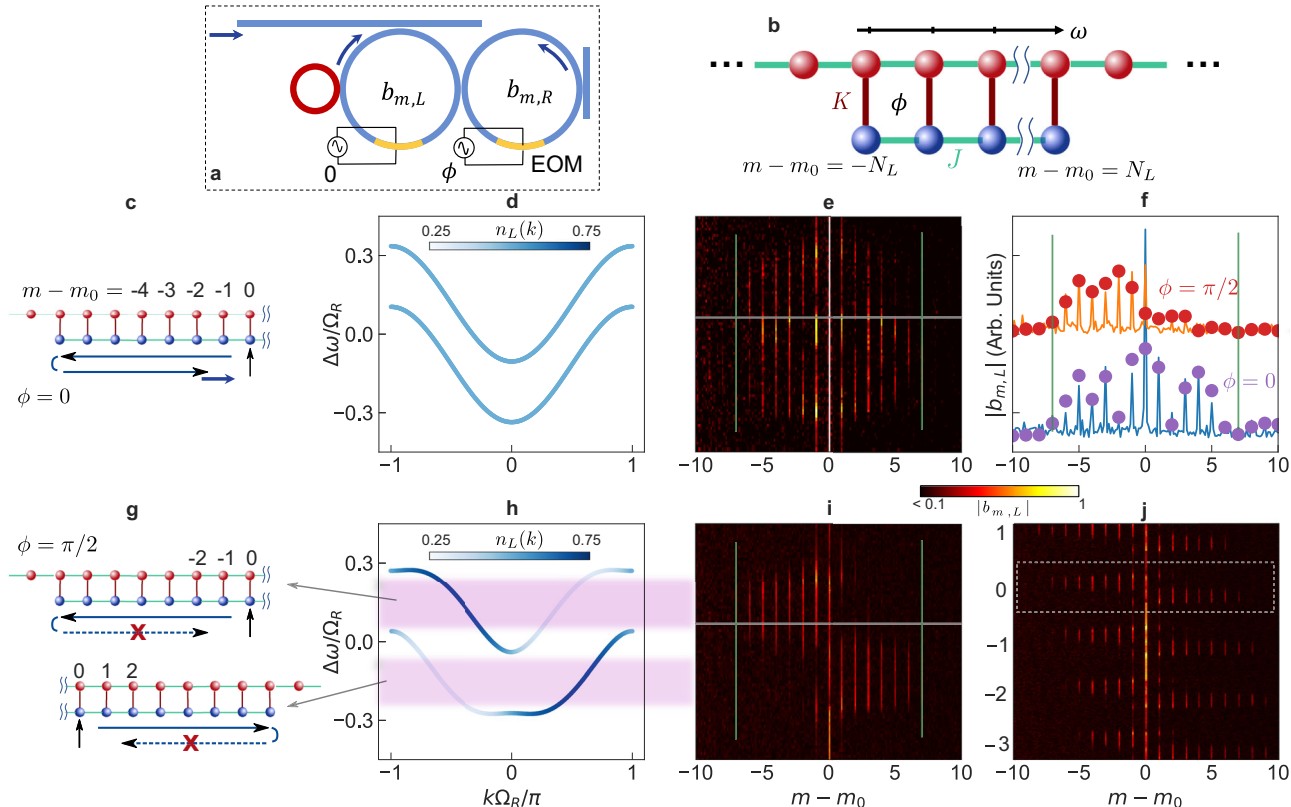

**Fig. 5 Interaction of quantum Hall ladder with boundaries. a** Two-leg quantum Hall ladder was constructed using two coupled rings, both of which are modulated with a phase difference $\phi$ between the two modulations. The auxiliary ring (red) introduces a boundary in the synthetic frequency dimension of the left ring, corresponding to a finite lower leg of the ladder in (**b**). **b** Shows a lattice model for (**a**). Coupling constants $J$ and $K$ are determined by the modulation amplitude in (**a**) and the evanescent coupling rate between the rings. **c** Schematic of the lattice excitation and its dynamics along the frequency dimension for the trivial case $\phi = 0$. The wave propagates along the frequency axis, reaches the boundary, and gets reflected, forming fringes in the steady-state intensity distribution. **d** Bulk band structure of the ladder for $\phi = 0$, showing symmetric bands with no chirality. **e** Measured lattice space occupation, showing fringes due to reflection from the boundary and bidirectional propagation. The fringes are clearly visible in the blue line cut in (**f**) taken at $\Delta\omega/\Omega_R = 0.11$. Purple dots in (**f**) represent Floquet scattering matrix simulation results. **g–i** Same as (**c–e**) but for the nontrivial topology case $\phi = \pi/2$. Chiral one-way modes are visible in (**h**) [colorbar represents the strength of localization on the ladder's lower leg, $n_L(k) = |b_L(k)|^2$]. Pink shaded regions depict energies with one-way modes, as verified experimentally in (**i**). Back-reflection is inhibited (schematics in (**g**)), leading to unidirectional propagation and no fringes despite the presence of a boundary, as visible in the orange line cut in (**f**). Red dots are simulation results. **j** Same as **i** but over a larger range of input laser detuning, showing the steady state for exciting various frequency sites. The boxed region corresponds to (**i**), near $\Delta\omega = 0$. For $\Delta\omega/\Omega_R$ near $\{-3, -2, -1, 1\}$, the input mode $m_0$ is also shifted by the same amount. y-axis labels in (**e**) and (**i**) represent $\Delta\omega/\Omega_R$ and are shared with (**d**) and (**h**).

We repeat the same measurement in the presence of the auxiliary ring (Fig. 4b). We see strong excitation of the system only at a discrete set of frequencies, as expected since the presence of the two boundaries results in a discrete set of eigenstates. For each of these eigenstates, the wavevector components spread over a range, centered at approximately where the wavevector would be at the same frequency for the infinite system. The experimental results in Fig. 4b agree excellently with numerical simulation results shown in Fig. 4c based on a Floquet scattering matrix analysis of the coupled ring system. Moreover, the numerical results indicate that the discrete eigenfrequencies that we observe in Fig. 4b agree with tight-binding simulations (Fig. 4d) where open boundaries are imposed on the two ends of a finite lattice, providing further evidence of a sharp boundary that we create.

**Demonstration of boundary effects in a quantum Hall ladder.**
We now demonstrate the effect of boundary on a topologically nontrivial system, the two-leg quantum Hall ladder[47], and show how it enables us to observe topologically robust transport of light along the frequency axis. To construct a two-leg quantum Hall ladder, we use a setup schematically shown in Fig. 5a, where we couple a pair of main ring resonators. The main ring on the left is

in addition coupled to an auxiliary ring. We ensure that the FSR of the main ring on the right matches the FSR of the coupled system consisting of the main ring on the left together with the auxiliary ring. We modulate both of the main rings at a frequency $\Omega_M = 2\pi \cdot 5.28$ MHz, which matches the FSR, with a relative phase difference $\phi$ in the modulations on the two rings[5]. The resulting Hamiltonian then describes a two-leg quantum Hall ladder[4,27,47] (Fig. 5b):

$$
\begin{aligned}
H_2 = &J \sum_{m=-N_L}^{N_L} b_{m,L}^\dagger b_{m+1,L} + J \sum_{m=-N_R}^{N_R} b_{m,R}^\dagger b_{m+1,R}\, e^{-i\phi} \\
&+ K \sum_{m=-N_L}^{N_L} b_{m,L}^\dagger b_{m,R} + \text{H.c.}
\end{aligned}
\tag{2}
$$

where $N_L$ and $N_R$ represent the number of frequency modes in the left and right legs of the ladder respectively, and $N_L < N_R$ due to the presence of the auxiliary ring that couples to the main ring on the left. $J$ is the modulation-induced hopping along the synthetic frequency dimension. $K$ represents the coupling between the two legs of the ladder, determined by the splitting ratio of the directional coupler that couples the two main rings together. The model in Eq. (2) exhibits a uniform effective magnetic flux $\phi$

permeating each square plaquette of the lattice. For $\phi \neq 0, \pi$, time-reversal symmetry is broken; such a model then supports one-way chiral states on each leg which are immune to back reflections from the boundary or corner (Fig. 5g, h). This one-way nature derives from a parent 2D quantum Hall insulator which manifests strong topological protection[47,48]. Thus, the setup allows us to study the interaction of boundaries with the topologically protected one-way chiral modes in a quantum Hall ladder.

To demonstrate the effect of the boundary as induced by the auxiliary ring, we excite the left main ring in the setup as shown in Fig. 5a. We choose the excitation frequency to match one of the lattice sites away from the boundary (Fig. 5c and g). In the case of $\phi = 0$, the band structure for an infinite two-leg system is shown in Fig. 5d. Since the system has time-reversal symmetry, the eigenstates equally occupy the left and the right legs and the system does not exhibit any chiral behavior. Consequently, with the excitation as shown in Fig. 5c, we expect that the generated field will propagate to both sides of the excitation site. Also, we expect to see interference fringes between the site of excitation and the boundaries. In Fig. 5e, we show the experimental results for this case where we measure the spectrum of the transmitted light via heterodyne detection (see Methods). We indeed observe that the output field contains strong components on both sides of the excitation site $m = m_0$. In Fig. 5f, we plot the amplitude at various lattice sites for $\Delta\omega/\Omega_R = 0.11$. We observe interference fringes due to the presence of the boundaries (indicated by green vertical lines), as exemplified by the dips at $m - m_0 = \pm 2$.

In the case of $\phi = \pi/2$, the band structure for the infinite system is shown in Fig. 5h. Since the system breaks time-reversal symmetry, the eigenstates show asymmetry in occupation between the left leg and the right leg, as illustrated in Fig. 5h where the color gradient shows the projection of the eigenstate on the left leg. Hence, with the excitation shown in Fig. 5g where the left leg is excited, we expect that the generated field will propagate to higher frequencies for the lower band, and to lower frequencies for the upper band, as determined by the sign of the group velocities of the chiral modes in Fig. 5h. Also, we do not expect to see interference fringes between the site of excitation and the boundaries, since the one-way nature of the chiral modes should suppress back reflection from the boundaries (schematics in Fig. 5g). In Fig. 5i, we show the experimental results for this case where we measure the spectrum of the transmitted light via heterodyne detection. Strikingly different from Fig. 5e, we indeed observe that the output field contains frequency components almost exclusively for modes to the left of the excitation ($m - m_0 \leq 0$) for the upper band, in the one-way detuning range shaded in pink in Fig. 5h. The direction of frequency conversion switches for the lower band. In Fig. 5f, we plot the experimentally measured amplitude at various lattice sites as the orange curve, which agrees well with Floquet scattering matrix simulations (red dots). The one-way nature, as well as the absence of interference fringes, are borne out in this amplitude distribution in frequency lattice space. Fig. 5j plots the amplitude distribution for a wide range of detuning $\Delta\omega$, corresponding to the excitation of different lattice sites $m_0$ along the frequency dimension. We observe that the topological robustness of light transport, as evidenced by the one-way nature and the lack of fringes, persists as we excite modes with different distances from the boundary. Note that the persistence of one-way propagation in the two-leg ladder limit attests to the topological robustness of the full 2D quantum Hall lattice independent of the boundary along the frequency axis. This is because the ladder preserves the modal structure of the edge states of the full 2D lattice in spite of the removal of all the bulk sites from the full 2D lattice, as predicted theoretically in ref. [47].

## Discussion

We have demonstrated the construction of sharp boundaries in synthetic dimensions by coupling an auxiliary ring resonator to a dynamically modulated ring, using a platform based on optical fibers. Recent progress in nanophotonic electro-optic modulators[49,50] incorporated into low-loss microring resonators provide opportunities for scalable on-chip integration of such concepts. This approach can be generalized to higher dimensions for exploring nontrivial topological boundary phenomena[25,51,52], both in conventional topological insulators as well as in higher-order topological insulators. While our demonstrations were limited to the simplest case of nearest-neighbor coupling, there are several ways to create boundaries in the presence of long-range coupling[28,32], a feature that is readily accessible in synthetic frequency dimensions. Examples include: (i) using multiple incommensurate rings, (ii) using perturbations to the cross section of the ring (as recently demonstrated in ref. [53,54]), and (iii) using dispersion engineering of the waveguide that comprises the ring. Our results also show that the energy of a synthetic lattice can be confined to a finite number of sites by coupling to additional auxiliary resonators, which is critical in efficient implementations of linear transformations or matrix-vector multiplications[21]. Our work should significantly advance the capabilities of synthetic dimensions in both topological photonics and for quantum[55] and classical signal processing.

## Methods

**Experimental details**. In this section, we provide a detailed description of the experimental setup corresponding to Fig. 5 of the main text. The setups for Figs. 2–4 can be obtained by disconnecting the second main ring on the right in Fig. 5a. We use a fiber ring resonator[32,46], with a lithium niobate phase modulator in each ring as the electro-optic modulator (EOM). The rings are excited by a low-noise continuous-wave laser (RIO Orion)[56], with a narrow linewidth <3 kHz. The main rings have a length of $L_0 \approx 38.6$ m, corresponding to a free-spectral range (FSR) of 5.35 MHz. Both the main fiber rings are coupled to through and drop ports to enable an independent calibration of the FSR of each ring when the coupling between the two rings is absent. The FSRs of the two rings are passively equalized by measuring the FSR of each ring and adding extra lengths of fiber or free-space sections in the second ring to compensate for the difference. The auxiliary ring consists of a loop of fiber containing a fiber polarization controller. The electrical signals used to drive the two modulators are derived from the same field-programmable gate array (FPGA), to ensure phase synchronization over long timescales. The phase difference between them was precisely controlled in software, and could be varied across the entire range $[0, 2\pi]$. By contrast, when independent function generators were used to drive the two modulators, we observed a continuous drift in the phase offset. Hence it was important to use two modulation signals derived from the same FPGA clock. The modulation signals were amplified by RF amplifiers before driving the EOMs. Each of the main rings also had an erbium-doped fiber amplifier (EDFA) to compensate for the losses from the EOM and other components. The auxiliary ring resonator had no EDFA. The amplified spontaneous emission noise from the EDFA was filtered using a dense-wavelength division multiplexing (DWDM) filter with a passband of 26.5 GHz centered at 1542.12 nm.

To enable strong coupling between the two rings ($K\tau_p > 1$), where $\tau_p$ is the photon lifetime in the main ring resonators and $K$ is the coupling rate between the main rings, we use polarization-maintaining (PM) fiber components at the junction between the rings. This is because the polarization axes of the two rings set by the EOMs are otherwise independent. The PM sections are illustrated by the blue fibers in Supplementary Fig. S1. The splitting ratio of the inter-cavity coupler is 75:25, that of the input coupler is 95:5, and that of the auxiliary ring coupler is 60:40. The through port signal was monitored on a slow photodiode (e.g., Fig. 2), whereas the drop port signal was sent to a fast photodiode with a bandwidth of 5 GHz after optical amplification with a semiconductor optical amplifier (not shown). For band structure measurement (Fig. 4), the drop port signal was directly sent to the photodiode in this manner. For frequency lattice-space measurements (Figs. 3 and 5), the drop port output was first mixed with the output of an acousto-optic modulator (AOM) before sending to the fast photodiode. The AOM shifted a part of the input laser by the RF drive frequency of 500 MHz to enable heterodyne detection of the cavity output.

Comparison with active mode-locking: In our setup, the presence of both an EDFA and a modulator is similar to that of an actively mode-locked laser. However, a few important differences exist: (i) The setup is operated completely below the lasing threshold; (ii) The input is around the same wavelength as the

output in the 1550-nm band, as opposed to lasers where the pump is at a significantly shorter wavelength than the lasing output. In our experiments, the EDFA only plays the role of mitigating roundtrip losses to achieve a high effective finesse for the cavity. If one operates the EDFA at gain higher than the roundtrip loss, especially with an amplitude modulator, actively mode-locked pulses can be produced[57,58].

## Data availability

The data generated in this study have been deposited in the Zenodo database under accession code 10.5281/zenodo.6516650 (https://doi.org/10.5281/zenodo.6516650).

## Code availability

The codes used to process the data generated in this study have been deposited in the Zenodo database under accession code 10.5281/zenodo.6516650 (https://doi.org/10.5281/zenodo.6516650).

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

## Acknowledgements
We acknowledge David A. B. Miller for providing lab space and equipment, and Meir Orenstein and Momchil Minkov for useful discussions. This work is supported by a Vannevar Bush Faculty Fellowship from the U.S. Department of Defense (Grant No. N00014-17-1-3030) and a MURI project from the U.S. Air Force Office of Scientific Research (Grant No. FA9550-18-1-0379). L.Y. also acknowledges the support of the National Natural Science Foundation of China (11974245, 12122407).

## Author contributions
A.D., L.Y., and S.F. conceived the project. A.D. carried out the experiments with assistance from K.Y.Y. and K.W. S.B. and A.D. developed the Floquet simulations. A.D., L.Y., K.Y.Y., K.W., J.V., and S.F. discussed the results and contributed to writing the manuscript. S.F. supervised the project.

## Competing interests
The authors declare no competing interests.
