## [Peer review file · Nature Communications]

REVIEWER COMMENTS

Reviewer #1 (Remarks to the Author):

The manuscript is a theoretical proposal and experimental demonstration of realizing synthetic frequency dimensions with specially introduced boundaries. The idea is to introduce an additional resonator, such that due to coupling some of the spectral lines are selectively detuned, and cannot be excited through a periodic modulation. The practical demonstration of this approach is very important, opening up the exploration of topological boundary effects. Specifically, the authors demonstrate topologically robust transport. These results would be of interest to the Nature Communications readers, and the manuscript can be recommended for publication after consideration of several comments:

- 1) The robustness of light transport is claimed several times in the manuscript. Could the authors elaborate more directly, which physical effects can disrupt the light transport in the usual (non-topological) regime, while the transport persists in the topological synthetic structures? In real space, such as photonic crystals, such an effect can be due to inhomogeneity in space. Are there inhomogeneities in synthetic space, or other factors?
- 2) In continuation of the previous question, how robust is the system to the fluctuation of the (small) differences in the rings, including the free-space section? Do topological features help here?
- 3) In other papers by this group, nonlocal couplings were considered. A discussion and outlook on the possibility of introducing boundaries under such a more general synthetic coupling case would be useful to include.

Reviewer #2 (Remarks to the Author):

The creation and applications of synthetic dimensions in driven systems has been a very active topic of research in condensed matter and quantum optics communities. The particular implementation that Authors consider in the present paper utilizes resonant modes of a ring resonator. In the absence of dispersion, internode transitions can be induced by a parametric drive at an integer multiple of the

mode frequency splitting (FSR). Theoretically, this maps to a tight binding lattice model. In the absence of dispersion and neglecting the existence of zero-frequency boundary, FSR is constant and this synthetic space is essentially unbounded. Thus injecting photons at a frequency at or near one of the modes (lattice “site”), leads to an unbounded spread through synthetic lattice, making it difficult to observe particularly interesting topological phenomena, which are most prominent at the “sample” edges.

What authors proposed and experimentally implement in this paper is an elegant solution — they couple the primary ring resonator to another one with a large FSR, so that the mode interaction “knocks” out some of the lattice sites, breaking it into segments that are not connected by the nearest neighbor hopping (provided by the parametric drive). They show that the resulting dispersion curve that they measure breaks up into discrete points, as expected for finite length segments. Moreover, by creating a two-leg quantum Hall system, they can clearly identify the edge propagation, which is the hallmark of Quantum Hall effect.

The work seamlessly combines experiment and theory and modeling, and in my opinion fully deserves publication in Nature Communications.

One question that I have, which is perhaps relevant to the current paper and the previous one where the synthetic band structure was measured, Ref [9], is about a relationship with mode-locking phenomena. The modulation that is being applied appears to be the same type of modulation that leads to formation of pulses in the mode locked lasers (see also Appendix B of <https://doi.org/10.1016/j.aop.2019.03.017>), with the input drive being similar to the laser pump. Yet, the output field in the present case is only sinusoidally modulated. Could authors explain the distinction between these two cases?

With this optional clarification I strongly recommend paper for publication.

Reviewer #3 (Remarks to the Author):

This manuscript presents the results of an experimental investigation in the area of synthetic dimensions in photonics where the synthetic degree of freedom is frequency. Previous work by this group and others have established that the use of active modulators in time can transfer optical signals across a regular frequency grid creating a discrete synthetic dimension, but this grid is in principle unbounded. In practice the grid has

a very large extent which prevents the investigation of the edge effects which are some of the most interesting aspects of topological and synthetic dimension systems.

As the frequency dimension is highly tunable and flexible, resolving this issue is a significant issue.

In this work, the frequency grid which is induced by a fiber ring resonator is modified by coupling to an auxiliary ring of higher free spectral range. Where the two frequency grids overlap, a splitting of the primary resonances occurs which prevents frequency hopping beyond that point. The system is thus restructured into a series of distinct finite frequency grids with clear boundaries. The paper presents several steps, showing first the alteration of the spectrum at the boundary points.

The fact of the boundary points is clearly demonstrated by the appearance of interference fringes in frequency space (an effect incidentally which would make for an elegant addition to an advanced undergraduate optics course).

The frequency limits also manifest in the observation of discretisation of the observed band structure.

Finally the paper demonstrates how the finite frequency lattice allows observation of non-reciprocal edge effects, analogous to those now familiar in spatially discrete topological systems. By controlling the phase difference between the driving fields of the two modulators the motion in frequency space can be made unidirectional or bidirectional with the presence or absence of the fringes mentioned earlier as a signature. All the experimental results are compared to calculations in idealised models based on discrete scattering and Floquet pictures with close agreement.

The paper is clear, convincing and interesting and given the prominence of this topic seems to me to be well worthy of publication in Nature Comm.

Despite my best efforts, I have only been able to identify two small suggestions for improvement:

- the theory approaches mentioned above are familiar techniques but no details on their application is provided. For readers new to this area, either more careful citation of the techniques or a few pages added to the supplementary information would be welcome.

- In Fig. 5, no y-axis scale or label is provided for subfigs e, i and j.

I suspect that for e and i, the scales and labels are shared with d and h, and for j, the

labels are shared, but as the x-axes are in different spaces I was unsure. Better to make this explicit I think.

Congratulations to the authors on a fine piece of work.

We thank all the reviewers for their comments and suggestions. Below we report our responses in green and indicate the modifications we have incorporated in our manuscript to address them.

Reviewer #1

The manuscript is a theoretical proposal and experimental demonstration of realizing synthetic frequency dimensions with specially introduced boundaries. The idea is to introduce an additional resonator, such that due to coupling some of the spectral lines are selectively detuned, and cannot be excited through a periodic modulation. The practical demonstration of this approach is very important, opening up the exploration of topological boundary effects. Specifically, the authors demonstrate topologically robust transport. These results would be of interest to the Nature Communications readers, and the manuscript can be recommended for publication after consideration of several comments:

We thank the reviewer for recognizing the importance of our work and the interest it would generate among Nature Communications readers.

1) The robustness of light transport is claimed several times in the manuscript. Could the authors elaborate more directly, which physical effects can disrupt the light transport in the usual (non-topological) regime, while the transport persists in the topological synthetic structures? In real space, such as photonic crystals, such an effect can be due to inhomogeneity in space. Are there inhomogeneities in synthetic space, or other factors?

2) In continuation of the previous question, how robust is the system to the fluctuation of the (small) differences in the rings, including the free-space section? Do topological features help here?

The reviewer points out an important aspect of synthetic dimension lattices that is in stark contrast to real-space lattices: the spatial inhomogeneity in e.g. photonic crystals is almost absent in frequency lattices. This has been estimated theoretically based on group velocity dispersion (see Supplementary information of our earlier paper [Dutt et al. Nature Communications 10, 3122 (2019)]). It has also been demonstrated experimentally in that paper as well as in papers by other groups [Zhang, Buscaino et al. Nature 568, 373 (2019); Yu, Reimer et al. Optica 7, 1189 (2020)], where hundreds of modes have been coupled using a single-frequency modulation. This attests to the equally spaced nature of the frequency modes over a very large range – an aspect that we allude to after Eq. (1) in our current manuscript. In short, an interesting aspect of synthetic dimensions is that we get a fairly uniform system with weak intrinsic disorder.

Since the standard intrinsic disorder is weak in synthetic dimensions, we demonstrate robustness through two signatures: (i) by showing the absence of back-reflections from a boundary in the topological phase, (ii) by showing that the one-way nature and chiral currents of the topological modes persist even in the two-leg ladder limit. The two-leg ladder is obtained from the full 2D quantum Hall insulator through a very strong perturbation, corresponding to the entirety of the bulk sites being removed, leaving behind a strip with just the edges. Both these effects are unique to the topologically non-trivial phase and does not occur in the trivial phase.

In our original manuscript, we highlighted the first signature of the absence of back-reflections from a boundary. In the revised manuscript, we also emphasize the one-way nature of the modes

and how it is related to the topological robustness of light transport by adding the following sentences before the Discussion:

“Note that the persistence of one-way propagation in the two-leg ladder limit attests to the topological robustness of the full 2D quantum Hall lattice independent of the boundary along the frequency axis. This is because the ladder preserves the modal structure of the edge states of the full 2D lattice in spite of the removal of all the bulk sites from the full 2D lattice, as predicted theoretically in Hugel and Paredes [PRA 89, 023619 (2014)].”

Regarding the reviewer’s concern on the fluctuations of ring-length differences, these fluctuations do affect the transport of light. This is true both for the topological phase as well as the trivial phase, and for the free-space section as well as the fiber section. Intuitively, a difference in the ring lengths would make the frequency modes drift from each other. If the frequency drift is more than approximately coupling rate between the rings, the equivalent lattice picture in Fig. 5 breaks down. In the supplementary information, we add a new section “Sensitivity to ring-length differences” where we show how small differences in the ring length affect light transport in this lattice using Floquet scattering matrix simulations.

3) In other papers by this group, nonlocal couplings were considered. A discussion and outlook on the possibility of introducing boundaries under such a more general synthetic coupling case would be useful to include.

In the revised manuscript’s discussion, we describe how boundaries can be introduced for a more general nonlocal couplings along the synthetic dimension:

“While our demonstrations were limited to the simplest case of nearest-neighbor coupling, there are several ways to create boundaries in the presence of long-range coupling, a feature that is readily accessible in synthetic frequency dimensions. Examples include: (i) using multiple incommensurate rings, (ii) using perturbations to the cross section of the ring (as recently demonstrated in Ref. [X. Lu, A. McClung, and K. Srinivasan, Nat. Photon. 16, 66–71 (2022)], and (iii) using dispersion engineering of the waveguide that comprises the ring.”

Hence, it is certainly feasible to introduce boundaries when long-range coupling is incorporated.

Reviewer #2

The creation and applications of synthetic dimensions in driven systems has been a very active topic of research in condensed matter and quantum optics communities. The particular implementation that Authors consider in the present paper utilizes resonant modes of a ring resonator. In the absence of dispersion, internode transitions can be induced by a parametric drive at an integer multiple of the mode frequency splitting (FSR). Theoretically, this maps to a tight binding lattice model. In the absence of dispersion and neglecting the existence of zero-frequency boundary, FSR is constant and this synthetic space is essentially unbounded. Thus injecting photons at a frequency at or near one of the modes (lattice “site”), leads to an unbounded

spread through synthetic lattice, making it difficult to observe particularly interesting topological phenomena, which are most prominent at the “sample” edges.

What authors proposed and experimentally implement in this paper is an elegant solution — they couple the primary ring resonator to another one with a large FSR, so that the mode interaction “knocks” out some of the lattice sites, breaking it into segments that are not connected by the nearest neighbor hopping (provided by the parametric drive). They show that the resulting dispersion curve that they measure breaks up into discrete points, as expected for finite length segments. Moreover, by creating a two-leg quantum Hall system, they can clearly identify the edge propagation, which is the hallmark of Quantum Hall effect.

The work seamlessly combines experiment and theory and modeling, and in my opinion fully deserves publication in Nature Communications.

One question that I have, which is perhaps relevant to the current paper and the previous one where the synthetic band structure was measured, Ref [9], is about a relationship with mode-locking phenomena. The modulation that is being applied appears to be the same type of modulation that leads to formation of pulses in the mode locked lasers (see also Appendix B of <https://doi.org/10.1016/j.aop.2019.03.017>), with the input drive being similar to the laser pump. Yet, the output field in the present case is only sinusoidally modulated. Could authors explain the distinction between these two cases?

With this optional clarification I strongly recommend paper for publication.

We thank the reviewer for recognizing the elegance of our boundary creation technique.

While actively mode-locked lasers also use sinusoidal modulation of a cavity, the major difference is that the mode-locked laser is operated above the lasing threshold, whereas the synthetic frequency dimension experiments we report here operate below the lasing threshold. Consequently, intracavity gain do not play a role in our experiments beyond mitigating roundtrip losses, while they play a central role in mode-locked lasers. Our system is linear in the optical fields, whereas gain saturation nonlinearity is important in the behavior of mode-locked lasers.

Note that the output field is not just sinusoidally modulated for a fixed input laser detuning, but a periodic pulse train with the periodicity equal to the cavity FSR. This was illustrated in Ref [9]. The band structure appears as one vertically stacks these pulse traces in the time domain for various detunings, and this band structure is sinusoidal for nearest-neighbor coupling. The output contains a large number of frequency components with a roughly exponentially decaying tail, whereas an actively mode-locked laser typically has a Gaussian profile.

The reviewer is correct that mode-locked lasers use a pump, but the pump is at a significantly shorter wavelength than the laser output (e.g. 980 nm or 1480 nm for a 1550 nm erbium-doped fiber laser). Synthetic frequency dimensions experiments use an input laser around the wavelength of detection (here in the 1550-nm band itself). In fact, our system is closer to electro-optic frequency combs, which also produce pulsed output, than to actively mode-locked lasers. If we use amplitude modulation instead of phase modulation and operate above the lasing threshold, the system then operates in the regime of actively mode-locked lasers. These

comparisons are briefly discussed in the paper referred to by the reviewer (Martin, *Annals of Physics* 405, 101 (2019) <https://doi.org/10.1016/j.aop.2019.03.017>), and as discussed in one of our recent papers (Yuan et al. *APL Photonics* 3, 086103 (2018)).

In the revised supplementary information, we add a paragraph to the “Experimental Details” section to discuss these ideas, since the most probable reason why this question could arise is the presence of both a gain element (EDFA) and a modulator in the setup:

“Comparison with active mode-locking: In our setup, the presence of both an EDFA and a modulator is similar to that of an actively mode-locked laser. However, a few important differences exist: (i) Our setup is operated completely below the lasing threshold; (ii) The input is around the same wavelength as the output in the 1550-nm band, as opposed to lasers where the pump is at a significantly shorter wavelength than the lasing output. In our experiments, the EDFA only plays the role of mitigating roundtrip losses to achieve a high effective finesse for the cavity. If one operates the EDFA at a higher gain such that lasing threshold is attained, especially combined with an amplitude modulator, actively mode-locked pulses can be produced [Martin, *Annals of Physics* (2019), Yuan et al. *APL Photonics* (2018)].”

Reviewer #3 (Remarks to the Author):

This manuscript presents the results of an experimental investigation in the area of synthetic dimensions in photonics where the synthetic degree of freedom is frequency. Previous work by this group and others have established that the use of active modulators in time can transfer optical signals across a regular frequency grid creating a discrete synthetic dimension, but this grid is in principle unbounded. In practice the grid has a very large extent which prevents the investigation of the edge effects which are some of the most interesting aspects of topological and synthetic dimension systems. As the frequency dimension is highly tunable and flexible, resolving this issue is a significant issue.

In this work, the frequency grid which is induced by a fiber ring resonator is modified by coupling to an auxiliary ring of higher free spectral range. Where the two frequency grids overlap, a splitting of the primary resonances occurs which prevents frequency hopping beyond that point. The system is thus restructured into a series of distinct finite frequency grids with clear boundaries. The paper presents several steps, showing first the alteration of the spectrum at the boundary points.

The fact of the boundary points is clearly demonstrated by the appearance of interference fringes in frequency space (an effect incidentally which would make for an elegant addition to an advanced undergraduate optics course). The frequency limits also manifest in the observation of discretisation of the observed band structure.

Finally the paper demonstrates how the finite frequency lattice allows observation of non-reciprocal edge effects, analogous to those now familiar in spatially discrete topological systems. By controlling the phase difference between the driving fields of the two modulators the motion in

frequency space can be made unidirectional or bidirectional with the presence of absence of the fringes mentioned earlier as a signature. All the experimental results are compared to calculations in idealised models based on discrete scattering and Floquet pictures with close agreement.

The paper is clear, convincing and interesting and given the prominence of this topic seems to me to be well worthy of publication in Nature Comm.

Despite my best efforts, I have only been able to identify two small suggestions for improvement:
- the theory approaches mentioned above are familiar techniques but no details on their application is provided. For readers new to this area, either more careful citation of the techniques or a few pages added to the supplementary information would be welcome.

- In Fig. 5, no y-axis scale or label is provided for subfigs e, i and j.
I suspect that for e and i, the scales and labels are shared with d and h, and for j, the labels are shared, but as the x-axes are in different spaces I was unsure. Better to make this explicit I think.

Congratulations to the authors on a fine piece of work.

We thank the referee for his/her favorable comments on our work.

In the revised manuscript, we carefully cite the following specific references where the theory approaches are discussed in significant detail, and introduce a section in the supplementary information for this purpose titled "Theory and simulation approaches":

- (i) A recent tutorial article: Yuan, Dutt, Fan, APL Photonics 6, 071102 (2021).
- (ii) The supplementary information of our recent papers: Science 367, 59 (2020), Nature Comm. 12, 2401 (2021), Leefmans et al. Nature Physics (2022) Section.

Regarding the reviewer's comment on Fig. 5 axis labels, we add an explicit clarification to the caption of Fig. 5 based on their suggestion: "Note that the y-axis labels in (e) and (i) represent detuning $\Delta\omega/\Omega_R$ and are shared with (d) and (h)."

REVIEWERS' COMMENTS

Reviewer #1 (Remarks to the Author):

The revision completely address my previous comments, and the paper is recommended for publication in the current form.

Reviewer #2 (Remarks to the Author):

Author have responded to referee comments. I recommend publication.

Reviewer #1 (Remarks to the Author):

The revision completely address my previous comments, and the paper is recommended for publication in the current form.

Response

We thank the reviewer for their positive recommendation.

Reviewer #2 (Remarks to the Author):

Author have responded to referee comments. I recommend publication.

Response

We thank the reviewer for their positive recommendation.